# Task-residual effective connectivity of motor network in transient ischemic attack

Truc Chu[1,2,7], Seonjin Lee[1,2,7], Il-Young Jung[3,7], Youngkyu Song[4], Hyun-Ah Kim[5], Jong Wook Shin [6✉] & Sungho Tak [1,2✉]

Transient ischemic attack (TIA) is a temporary episode of neurological dysfunction that results from focal brain ischemia. Although TIA symptoms are quickly resolved, patients with TIA have a high risk of stroke and persistent impairments in multiple domains of cognitive and motor functions. In this study, using spectral dynamic causal modeling, we investigate the changes in task-residual effective connectivity of patients with TIA during fist-closing movements. 28 healthy participants and 15 age-matched patients with TIA undergo functional magnetic resonance imaging at 7T. Here we show that during visually cued motor movement, patients with TIA have significantly higher effective connectivity toward the ipsilateral primary motor cortex and lower connectivity to the supplementary motor area than healthy controls. Our results imply that TIA patients have aberrant connections among motor regions, and these changes may reflect the decreased efficiency of primary motor function and disrupted control of voluntary movement in patients with TIA.

[1] Research Center for Bioconvergence Analysis, Korea Basic Science Institute, Cheongju 28119, Republic of Korea. [2] Graduate School of Analytical Science and Technology, Chungnam National University, Daejeon 34134, Republic of Korea. [3] Department of Rehabilitation Medicine, Chungnam National University Sejong Hospital, Sejong 30099, Republic of Korea. [4] Bio-Chemical Analysis Team, Korea Basic Science Institute, Cheongju 28119, Republic of Korea. [5] Department of Rehabilitation Medicine, Chungnam National University Hospital, Daejeon 35015, Republic of Korea. [6] Department of Neurology, Chungnam National University Sejong Hospital, Sejong 30099, Republic of Korea. [7] These authors contributed equally: Truc Chu, Seonjin Lee, Il-Young Jung. ✉email: dr.shin@hanmail.net; stak@kbsi.re.kr

Transient ischemic attack (TIA) is classically defined as focal brain ischemia with transient neurological deficits that resolve within 24 h after onset[1,2]. Typical TIA symptoms involve hemibody numbness, hemiparesis, aphasia, dysarthria, diplopia, facial weakness, ataxia, and vertigo[3]. TIA symptoms can be categorized into motor, sensory, and visual symptoms, where the motor symptoms include hemiparesis, unilateral arm/face/leg weakness, loss of motor function, and loss of muscle power[4]. Although the episodes of symptoms are transient, the risk of stroke occurring after a TIA is high[5,6]. According to a cohort study with 87 patients, the early stroke risk after TIA was 8.0% at 7 days, 11.5% at 1 month, and 17.3% at 3 months[6].

Previous studies showed impairments in the behavioral performance of patients with TIA, and they involved cognitive and motor functions[7–9]. For example, more than one-third of patients with TIA had impairment in at least one cognitive domain, including working memory, attention, and information processing, 3 months after the event[7]. Recently, with the application of a robotic exoskeleton system, it has been shown that patients with TIA also had impairments in motor and cognitive-motor function that can last for over 1 year[8,9]. Specifically, this robotic system allowed the use of assessment paradigms for quantifying motor functions (e.g., visually guided reaching), proprioception functions (e.g., arm position matching), and cognitive-motor functions (e.g., reverse-visually guided reaching). Nearly 51.3% of the patients were found to have an impairment on a given task two weeks after a TIA event, and up to 27.3% had an impairment after a year. Besides behavior performance, patients with TIA also experienced changes in brain structure. Studies showed that TIA resulted in delayed brain atrophy after 90 days[10] and increased the atrophy rate over 3 years[11]. Abnormalities in white matter[12,13] or gray matter tract[10,14] were also reported. These findings suggest there might be disruptions at many levels in the brain despite the absence of infarctions and the resolved symptoms.

Several abnormalities in the functional brain activity were revealed in patients with TIA using functional magnetic resonance imaging (fMRI). fMRI is a non-invasive method to measure the blood oxygenation level-dependent (BOLD) changes in the brain[15]. Using task-based fMRI, Su et al.[16] detected increased activations in the inferior frontal gyrus, dorsolateral prefrontal cortex, insula, inferior parietal lobe, and cerebellum during verbal working memory tasks one week after TIA. Using resting-state fMRI, a reduction in the amplitude of low-frequency fluctuation[17,18] was observed in the left cerebellum, posterior lobe, precentral/postcentral gyrus[19], and the middle temporal gyrus[20]. These results characterized the disruptions of regional functional changes during resting- and task-states in patients with TIA, compared with healthy participants.

Several studies found decreases in resting-state functional connectivity 1 month[21] and even 4 years[22] after TIA events. Functional connectivity is defined as the temporal correlation between spatially remote neurophysiological events[23]. Specifically, Li et al.[21] found decreased functional connectivity in the networks related to cognitive function (e.g., default mode network, dorsal attention network, central executive network, and core network). They also observed changes in the motor network. Indeed, patients with TIA exhibited declined functional connectivity in the somatomotor network, including the left postcentral gyrus. Together with the previous findings[19,21], these results imply that patients with TIA experience a decrease in spontaneous connectivity of motor-related regions during the resting state. However, to our knowledge, there have been no studies focusing on the investigation of task-dependent modulation of functional connectivity in patients with TIA and associating the difference in connectivity between TIA and healthy controls with the risk of stroke.

The main contribution of this study is to characterize the changes in task-residual effective connectivity of TIA during motor processing, compared with that of healthy controls. Effective connectivity is the model-based influence of one region on another and informs the directed causal interactions among brain regions underlying neuronal processes[23,24]. Our hypothesis is that effective connectivity patterns of patients with TIA will show alterations, specifically stronger positive connectivity toward the ipsilateral motor areas such as the primary motor and premotor cortices, compared to healthy controls. Although the ipsilateral motor regions are known to be inhibited during the movement of the hand in healthy young adults, previous studies have shown that these ipsilateral regions become less inhibited or receive excitatory influences from contralateral regions in older adults and people with motor disabilities such as stroke[25–27]. This can be caused by an effort to compensate for the reduced function or the decrease in neuronal efficiency[28]. Therefore, we expect that an excitatory connection towards ipsilateral motor regions would still remain in patients with TIA after removing aging effects from experimental data of TIA, and this aberrant connectivity would be associated with risk factors of stroke in TIA.

Regarding methods for this study, we extend spectral dynamic causal modeling (DCM)[29,30] to estimate task-residual effective connectivity and apply this to 7 T fMRI. Recently, a growing body of brain connectivity research has been using task-residual data, the remaining time series after removing task-related signals[31,32]. Having been applied to functional connectivity, task-residual connectivity was shown to be different from resting-state connectivity[31], and still reflects the connectivity between task-related regions[33–36]. For example, Al-Aidroos et al.[34] reported an increase in task-residual connectivity between the occipital cortex and the fusiform face area during attention to faces, while attention to scenes strengthened the connection between the occipital cortex and the parahippocampal place area. They also found an association between the attentional modulation in the connectivity and behavioral performance of the participants, suggesting that task-residual connectivity could reflect task engagement. Nevertheless, these task-residual neuronal fluctuations have not been exploited in effective connectivity. Therefore, in this study, we apply computationally efficient spectral DCM to task-residual time series of 7 T fMRI, and explore the changes in task-residual effective connectivity from patients with TIA.

## Results

**Characteristics and demographics of participants.** Table 1 shows the clinical characteristics and demographics of healthy controls (HC) and TIA participants. The two-sample $t$-test was applied to test for differences in ages between the two groups ($n_{HC} = 28$, $n_{TIA} = 15$). The results showed no significant differences between the two groups with respect to age (HC = 63.8 ± 6.6, TIA = 61.5 ± 9.5 years).

**Table 1 Characteristics and demographics of participants.**

|  | HC | TIA |
|---|---|---|
| Number of subjects | 28 | 15 |
| Age (years) | 63.8 ± 6.6 | 61.5 ± 9.5 |
| [a]Sex (male/female) | 8/20 | 11/4 |
| Handedness (right/left) | 28/0 | 15/0 |
| ABCD$^2$ score | — | 4.7 ± 0.9 |
| MRI scan date after the event | — | 16.2 ± 7.4 |

Values are the mean ± standard deviation. Two-sample $t$-test and Chi-square test were performed to test the differences in age and sex between healthy controls (HC) and patients with transient ischemic attack (TIA), respectively.
[a]Indicates statistically significant difference based on $p < 0.05$.

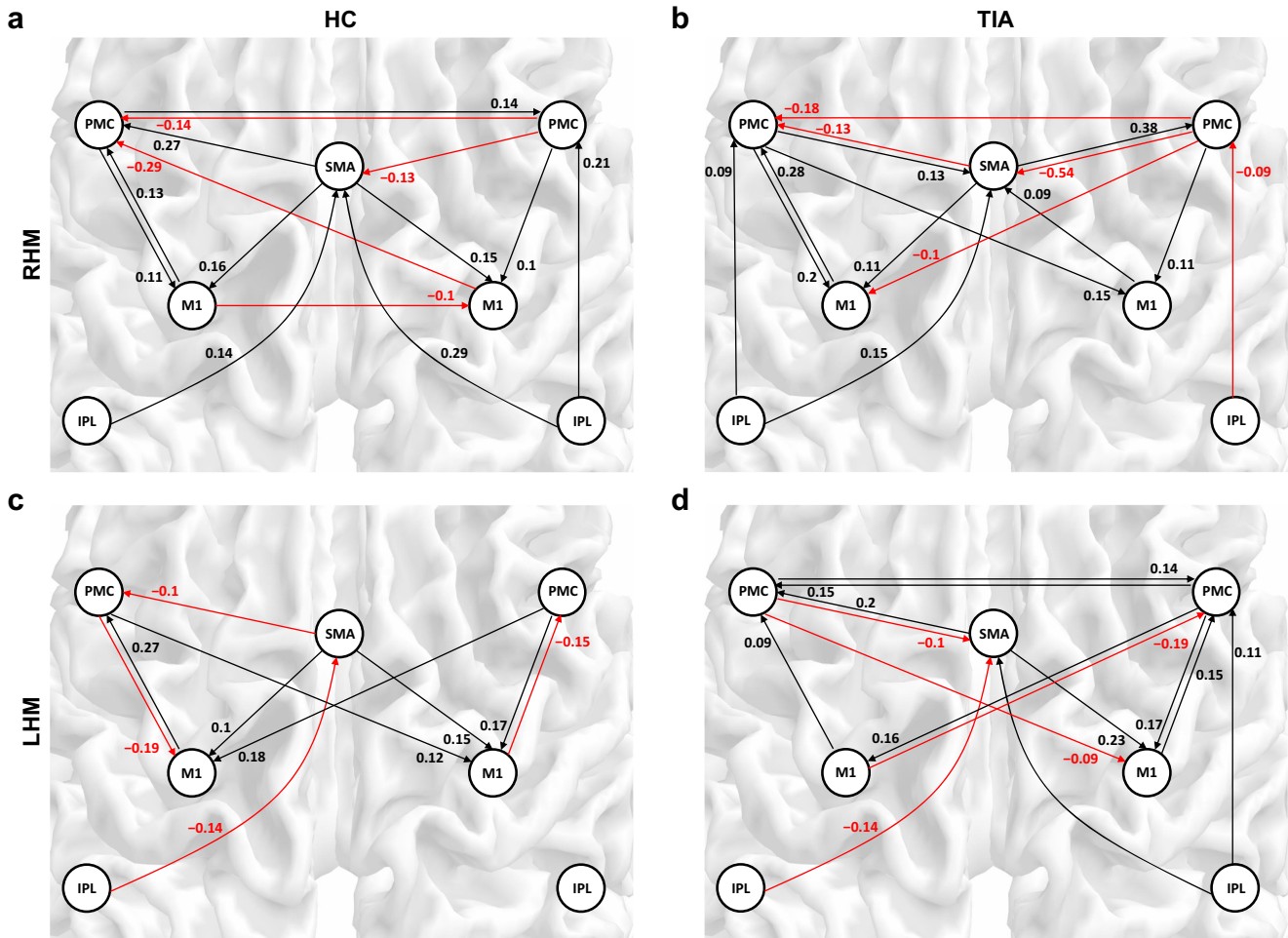

**Fig. 1 The posterior mean of the task-residual effective connectivity changes in the transient ischemic attack (TIA) and healthy control (HC) participants during fist-closing movements.** Black arrows represent positive (excitatory) effective connectivity, while red arrows represent negative (inhibitory) effective connectivity during **a**, **b** right-hand (RHM) and **c**, **d** left-hand movements (LHM). All the connections presented have significant evidence based on the Bayesian criterion of $p > 0.99$ ($n_{HC} = 28$, $n_{TIA} = 15$). For the numerical source data, see Supplementary Data 1. SMA supplementary motor area, PMC premotor cortex, M1 primary motor cortex, IPL inferior parietal lobule.

However, there was a significant difference in the sex ratio between the two groups ($p < 0.05$), according to the Chi-square test.

**Task-residual effective connectivity.** Figure 1 shows the posterior mean of the task-residual effective connectivity changes for the best model structure in the TIA and HC participants. All the connections represented had significant evidence based on the Bayesian criterion of $p > 0.99$. Both groups showed bilateral excitatory connections toward the primary motor cortex (M1) during right and left-hand movements. These results for healthy controls (mean age > 63 years) were in agreement with previous studies, which showed bilateral excitatory patterns of the motor network during hand movements in older adults[25,26]. Thus, these results also confirmed that although the task-induced canonical response was regressed out, trial-by-trial variability relevant to the task still remained in the residual signal, which led to similar patterns of task-residual effective connectivity with those of the previous results.

Figure 2a, b shows the task-residual effective connectivity having significant group differences in response to right and left-hand movements. Stronger excitatory connections from the left premotor cortex (PMC) to the ipsilateral M1 were observed in the patients with TIA than the HC, during both right-hand movement (from the left PMC to the right M1: 0.08) and

left-hand movement (from the left PMC to the left M1: 0.14). In addition, during right-hand movement, the patients with TIA had a stronger excitatory influence from the contralateral M1 to the ipsilateral M1 (0.08), compared to the HC. Therefore, although there were inhibitory effects from the supplementary motor area (SMA) to the ipsilateral M1 greater in the patients with TIA than the HC (−0.11 and −0.1 for the right and left-hand movements, respectively), the sum of all connections towards the ipsilateral M1 was positively increased in the patient with TIA, compared with the HC during both right and left-hand movements. Regarding the SMA region, the effective connectivity to the SMA was decreased in the patients with TIA, compared with the HC during right-hand movement (the right PMC to the SMA: −0.15, the right inferior parietal lobule (IPL) to the SMA: −0.13) and left-hand movement (the right M1 to the SMA: −0.08). Regarding the PMC regions, the patients with TIA had higher suppressive influence to the PMC during right-hand movement, while for the left hand movement higher excitatory effects to the PMC were observed from the patients with TIA than the HC. Individual connectivity strengths of each group, HC and TIA, are shown in Fig. 2c, d, respectively. The overall distribution of individual data points for each effective connectivity corresponds to their representative values—the group-level posterior mean of effective connectivity. However, please note that the average

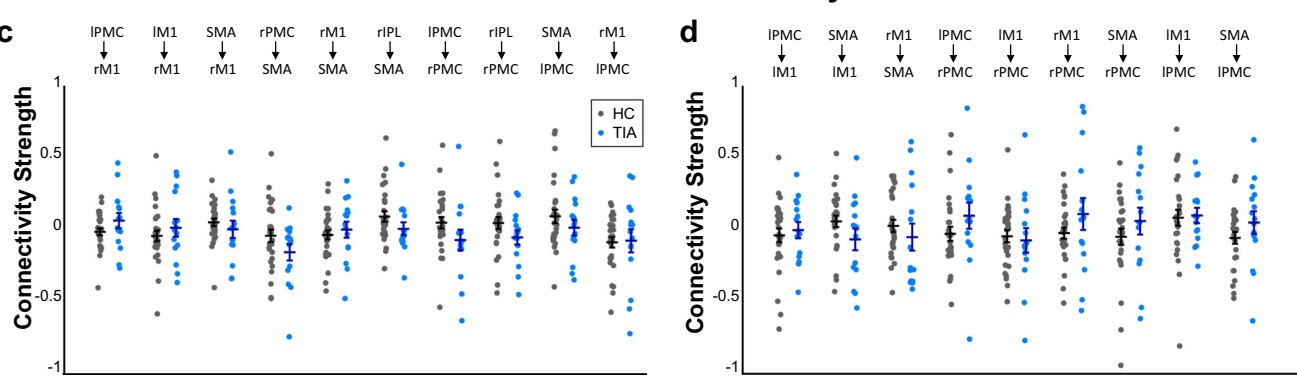

**Fig. 2 The significant group differences of the task-residual effective connectivity in response to fist-closing movements.** Black arrows represent stronger connectivity in the transient ischemic attack (TIA) group compared to healthy control (HC), while red arrows represent weaker connectivity in the TIA group compared to HC during **a** right-hand movement (RHM) and **b** left-hand movement (LHM). All the connections presented have significant evidence based on the Bayesian criterion of $p > 0.99$ ($n_{HC} = 28$, $n_{TIA} = 15$). **c, d** Individual connectivity strengths of each group, HC and TIA during RHM and LHM, respectively. Black and red dots indicate the individual level posterior mean of effective connectivity estimated from HC and TIA, respectively. Horizontal lines represent the average values of individual dots. Error bars represent the standard error of the mean. For the numerical source data, see Supplementary Data 1. SMA supplementary motor area, PMC premotor cortex, M1 primary motor cortex, IPL inferior parietal lobule.

values of the estimated connectivity strengths from individuals are not the same as the group-level parameters of the posterior mean. This discrepancy arises because the Parametric Empirical Bayes (PEB) framework used for the group analysis of this study considered both the expected values and the covariance of the individual-level parameters for estimating group-level parameters, whereas the simple average values of the individual data points would ignore the estimated covariance parameters[37].

Figure 3a, b shows the posterior mean of the task-residual effective connectivity of the external control regions of interest (ROIs) in the TIA and HC groups during right and left hands movements, respectively. Both groups of the TIA and HC had inhibitory connections within the default mode network during the task state. There was no significant group difference in the effective connectivity among these control ROIs between HC and patients with TIA. These results support that the connectivity difference between TIA and HC during fist-closing movements only arose in the task-specific motor-related regions.

## Discussion

In this study, we investigated the task-residual effective connectivity changes in the motor areas during fist-closing movement

in order to shed light on the underlying pathophysiology of TIA. We found that there were increased excitatory connections toward the ipsilateral M1 and decreased connections toward the SMA in patients with TIA compared to the HC group. These changes in connectivity were consistent during both right-hand and left-hand movements.

In the following, we interpret and discuss the significance of our findings in more detail. During fist-closing movements, the TIA and HC groups exhibited excitatory connections toward both contralateral and ipsilateral M1. The bilateral excitatory connections toward the M1, in other words, hemispheric asymmetry reduction, is in agreement with previous research in older adults[25,26]. Notably, a stronger excitatory connection toward the ipsilateral M1 was observed in the TIA group compared to the HC group. This indicated that in patients with TIA, the hemispheric asymmetry was further reduced. This is consistent with previous research on stroke patients, which showed less inhibition/stronger excitation connection toward the ipsilateral M1, when patients performed paretic hand movements[27].

To explain the reduction in hemispheric asymmetry, there are currently two ideas that are under debate. One of them is compensatory, which hypothesizes that the increase in recruitment of the ipsilateral hemisphere is responsible for supporting the motor

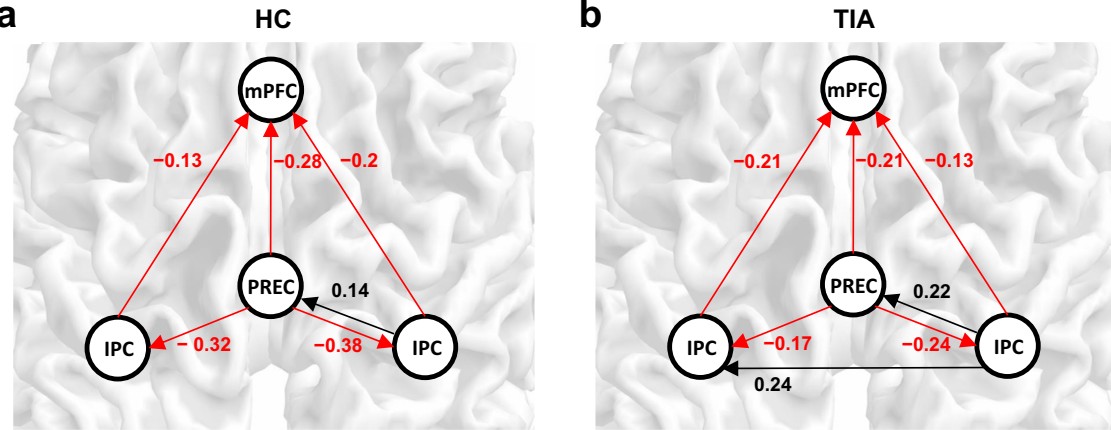

**Fig. 3 The posterior mean of the task-residual effective connectivity of the external control regions of interest (ROIs) in the transient ischemic attack (TIA) and healthy control (HC) groups during right and left hands movements. a** Connectivity estimated from HC, **b** Connectivity estimated from TIA. Black arrows represent positive (excitatory) effective connectivity, while red arrows represent negative (inhibitory) effective connectivity. All the connections represented had significant evidence based on the Bayesian criterion of $p > 0.99$ ($n_{HC} = 28$, $n_{TIA} = 15$). For the numerical source data, see Supplementary Data 2. mPMC medial prefrontal cortex, PREC precuneus, IPC inferior parietal cortex.

function of the contralateral hemisphere, whose functions had declined due to old age or stroke. This hypothesis implies that average activation of the ipsilateral hemisphere would be positively related to behavioral performance[28]. The alternative idea is that this non-selective activation or de-differentiation perhaps reflects neural inefficiency or motor disinhibition that occurred in the less precise brains of older adults[28,38]. Our results support the latter hypothesis (i.e., de-differentiation). In this study, we observed higher excitatory connections toward the ipsilateral M1 during motor processing in patients with TIA compare to HC. This could reflect a decrease in neural efficiency in these subjects.

Several studies have shown that the increase in excitatory connection toward the ipsilateral M1 was concomitant with the decrease in motor function and correlated with the increase in disease state (age, stroke severity, or stroke risk). Specifically, using transcranial magnetic stimulation and DCM, Boudrias et al.[25] have found that the increased excitatory influence of the contralateral M1 to ipsilateral M1 correlated with advancing age. In addition, a relationship was observed between higher recruitment of the ipsilateral sensorimotor cortex in older adults and longer reaction time, which reflects poorer motor performance[39]. Similar patterns were also observed in stroke patients, where the increase in activation of the ipsilateral M1 was found to be correlated with stroke impairment and poor motor performance[40,41]. Moreover, DCM analysis also showed the relationship between higher excitatory connectivity from the contralateral M1 toward ipsilateral M1 with stronger motor impairments and poorer motor recovery[41,42]. In agreement with the findings of these previous studies, we showed a positive significant correlation between higher excitatory connections toward the ipsilateral M1 and the ABCD[2] score—a higher stroke risk factor, which supports a de-differentiation mechanism in patients with TIA as a result of decreased neural efficiency. Details of the correlation analysis and results are reported in Supplementary Note 1 and Supplementary Fig. 1, respectively.

The connections from other regions to the SMA were found to be reduced in the patients with TIA compared to the HC group (from the right IPL and right PMC during right-hand movement, the right M1 during left-hand movement). The SMA is recruited in the temporal processing of the movement, the preparation and execution of voluntary movements, and is involved in the process of linking (external or internal) stimulus to actual movements[43–45]. The SMA is anatomically connected with the

M1[46]. In a disconnectome study of patients with stroke, structural disconnection induced by a lesion in the medial PMC, the SMA[47], was shown to be a contributor in the prediction of motor function impairments at 2 weeks and up to 1 year after strokes[48]. Because, as an experimental protocol of this study, the subjects were instructed to close and open their hands synchronized with a 1 Hz flickering circle (movement cue), the SMA was activated during the task and thus was involved in the connectivity model in line with previous studies[27,49–51].

We further validated whether the SMA plays an essential role in the fist-closing movement synchronized with the visual movement cue (i.e., a relatively simple visually guided movement process), by testing 2 models; a fully connected model and one without bidirectional connectivity to the SMA. Bayesian model selection based on random-effects analysis[52] was used to determine the most likely among the candidate models given the observed data. As shown in Fig. 4, the best model was the fully connected model with higher model exceedance probability for both right (Fig. 4a) and left fist-closing movements (Fig. 4b), compared to the model without connectivity to the SMA. This result indicates that the motor network requires the SMA in the process of the fist-closing movement synchronized with the visual movement cue.

In this study, we observed the connectivity towards the SMA was decreased in the patients with TIA compared to the healthy controls. As described in the introduction section, several studies have shown decreased behavioral performance of cognitive and motor functions in patients with TIA[8,9]. Using a robotic exoskeleton system, the patients with TIA were found to have an impairment in visually guided motor functions after a TIA event. Therefore, the reduced connectivity to the SMA in patients with TIA could reflect the reduced function of temporal processing during the visually guided fist-closing movement that can be inferred from typical symptoms of patients with TIA. Similar patterns of connectivity changes (i.e., decrease in connectivity to SMA) have been observed during visually cued fist-closing movements from the patients with chronic stroke[53] and older adults[25] known to have some difficulties in the control of movement, compared with healthy controls.

Similar to the SMA region, several effective connections toward the bilateral PMCs were found to be decreased in patients with TIA during right-hand movement (from the SMA to the left PMC, the left PMC to the right PMC, and the right IPL to the

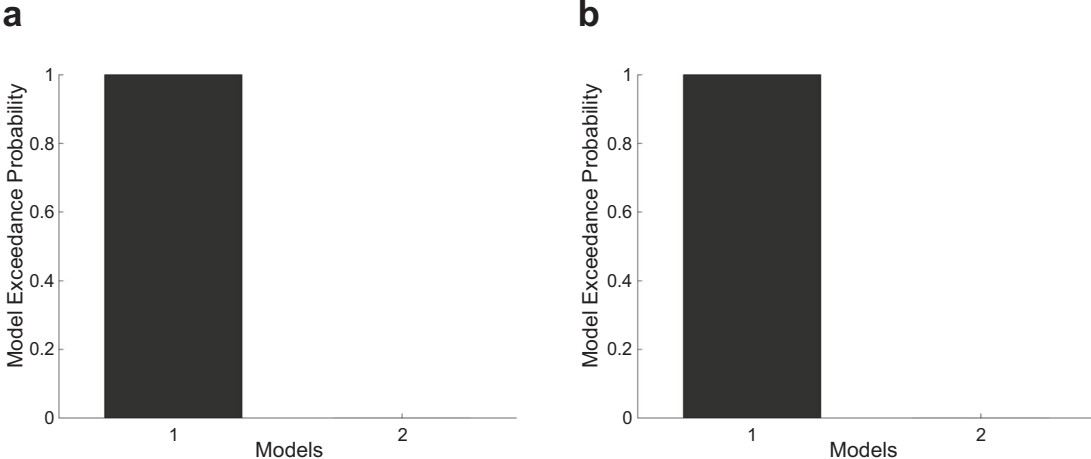

**Fig. 4 Results of Bayesian model selection based on random-effects analysis. a** Model exceedance probability of the fully connected model (Model 1) and model without bidirectional connection to the supplementary motor area (SMA) (Model 2) across subjects ($n = 43$) for right-hand movement (RHM) data. **b** Model exceedance probability for left-hand movement (LHM) data. For the numerical source data, see Supplementary Data 3.

right PMC), compared to the healthy controls. This effect was followed by increased connectivity from the PMC to the ipsilateral M1. The anatomical location of the PMC allows it to receive and process the information from the parietal cortex and project it to the M1 for motor processing. Functionally, the PMC plays roles in motor control such as directing goal-oriented actions, where there is a need to incorporate visual sensory information into motor actions[54]. Therefore, the decreased connectivity toward the PMC may reflect the decrease in the information processing of sensory stimuli to the actual movement in the patients with TIA, compared with healthy controls. In patients with stroke, Sharma et al.[55] have found a reduction in the effective connectivity between the SMA and PMC, during both motor imagery and motor execution of finger-thumb opposition movement tasks in stroke patients. Functional connectivity between the SMA and PMC was also found to be decreased in patients with Parkinson's disease[56]. On the other hand, during left-hand movement, greater connectivity to the PMC was observed from the patients with TIA than the HC. However, there was no significant difference in excitatory connections from the PMC to the contralateral M1 between the TIA and HC groups, which implies that these changes in connectivity were not followed by compensatory effects on the motor execution function in patients with TIA.

There are limitations to this study that need to be addressed. First, the group size in this study was relatively small. The numbers of HC and TIA participants were 28 and 15, respectively. Recent studies have identified the sample size as an influential part of the reproducibility problem, particularly in the behavior-brain activity relationship estimated by fMRI experiment[57–59]. Marek et al.[57] quantified this issue by demonstrating that cross-sectional brain-behavior correlations are unreliable without large samples. On the other hand, they also reported that neuroimaging-only studies are typically adequately powered at small sample sizes, and showed that central tendencies of resting-state functional connectivity can be reliably represented averaging within 25 samples[57]. These differences in the statistical power analysis result[57] were in line with an opinion of the study[58]: brain activity-behavior correlations inherently have lower power than average group-activation effects. The main findings of our study were based on the protocols of the neuroimaging-only studies.

In the neuroimaging-only studies, several papers have reported the sample size estimates required to achieve 80% power at a 5%

level of significance[60–62] and provided guidance on improving power by increasing the effect size[58,62–64]. The statistical power is defined as the probability of rejecting the null hypothesis given that the alternative hypothesis is true, and the effect size $\delta$ is typically calculated as the absolute difference of the means $\mu_1$, $\mu_2$ of the experimental conditions or independent groups divided by its standard deviation $\sigma$: $\delta = \frac{|\mu_2 - \mu_1|}{\sigma}$. For instance, Desmond and Glover[60] used empirical data acquired from 3 T MRI to calculate the effect size based on the percent signal change between conditions and the variability consisting of the intra- and inter-subject standard deviations. These estimated values were then used to create the power curves showing the relationship between the sample size and power. The results showed that 12 subjects are required to ensure 80% power at a single-voxel significance level ($\alpha = 0.05$) and approximately 25 subjects are needed at a more conservative level ($\alpha = 0.000002$). In terms of the DCM-fMRI study, Goulden et al.[61] performed the power analysis using the typical difference of the means of the effective connectivity between groups and the population standard deviation (the bootstrap distribution) of the parameters. They showed that for the emotion processing and n-back task data, 20 subjects per group are required to achieve 80% power at a significance level of 0.05.

Notably, given the statistical power and significance level, the minimum number of samples (subjects) $n$ can be further reduced by increasing the effect size $\delta$: $n \propto 1/\delta^2$. The effect size is improved by increasing the signal-to-nose ratio (SNR) of the acquired images[63], which can be achieved by using ultra-high field (e.g., 7 T) MRI system[62,65–67]. Specifically, compared to 3 T, fMRI at 7 T allowed for measuring the increased BOLD contrast in the same (motor) region, and increasing the $t$-statistic value and functional connectivity strength calculated over a common region of interest[65,66]. In terms of the DCM connectivity, the posterior entropy at 7 T was substantially less than that of 3 T estimates[67]. Using these gains at 7 T in the effect size, Torrisi et al. performed power analyses between field strengths and clearly showed that fewer subjects at 7 T are necessary to produce comparable effects at 3 T[62].

In this study, we estimated the DCM connectivity strengths from the 7 T fMRI images. Therefore, considering (i) the increase in effect sizes at 7 T and (ii) power analysis results of previous neuroimaging studies[57,60–62], the sample sizes used in this study ($n_{TIA} = 28, n_{HC} = 15$) fall within the reasonable range of the sample sizes to ensure 80% power at a 5% level of significance.

We also performed a power analysis using DCM connectivity data obtained from patients with stroke[27]. The experimental task and ROIs of the connectivity model in Rehme et al.[27] were similar to our study. In Rehme et al.[27], the mean difference between stroke patients and healthy subjects in effective connectivity among motor areas during fist-closing movements was 0.037. Standard deviations of effective connectivity for stroke and healthy subjects were 0.03 and 0.04, respectively. Using these data, we calculated the sample size required in each group $n_i$ for this study[68,69] and resulted in $n_i = 14.22$ to ensure 80% power $(1 - \beta)$ at a 5% significance level ($\alpha$): $n_i = 2(\frac{Z_{1-\alpha/2}+Z_{1-\beta}}{\delta})^2 = 2(\frac{1.96+0.84}{1.05})^2 = 14.22$. Therefore, we can confirm that the number of subjects acquired for each group in this study met the criterion for the minimum sample size that was approximated using power analysis with a similar task and patient group dataset. We only reported the posterior means of the connections that were above the Bayesian significance criterion of $p > 0.99$, to confirm the statistical significance of our results. Nevertheless, caution should be taken when generalizing the discussion particularly based on the relationship between connectivity strength and ABCD[2] score that would be required for a larger sample size.

Second, the sex ratio between the participants with TIA and HC was significantly different. Therefore, to remove the effects of sex difference between groups on the connectivity analysis in this study, we included the sex difference (1 for male, −1 for female) as a nuisance covariate in the 2nd-level general linear model (GLM) and DCM analysis. This strategy corresponds to the standard analysis methods described in Penny and Holmes[70]; Mumford and Nichols[71]; Zeidman et al.[37]. This approach was applied in previous studies at the 2nd-level analysis to remove the confounding effect of between group's differences factors (such as age, sex, or educational level) on the main finding of the studies[72–74]. For example, Nair et al.[73] have added sex as the covariate of no interest in group-level DCM analysis in a study of effective connectivity changes in Huntington's disease. Modinos et al.[72] have added age (which was different between groups) as the covariate of no interest in both group-level GLM and DCM analyses. Although the previous study[75] reported that female healthy adults had higher risk factors and severity of stroke than male healthy adults, a cohort study on TIA found that stroke recurrence, the composite outcome of stroke, myocardial infarction, and death risk showed no differences between genders[76]. Therefore, based on the analysis of this study and previous findings, the effect of sex difference was mitigated and would not bias the estimates of ROIs and task-residual effective connectivity among the ROIs.

In summary, as the main results, this study showed that while the patients with TIA performed fist-closing movements according to the visual stimuli of flickering circles (1 Hz), the patients with TIA had greater connections to the ipsilateral M1 and lower connections to the SMA and PMC than the healthy controls. These experimental results may reflect the potential disruptions in the information processing (visual cue to action) and controls of voluntary movement, and decreased efficiency of primary motor function M1 in patients with TIA. Previous TIA studies have shown changes in the brain activity during resting state[19,20] and working memory task[16], as well as decreased functional connectivity in the resting brain[21], in patients with TIA. Our studies added findings to this field by revealing the disruptions in the task-residual effective connectivity of the motor network in TIA during visually cued fist-closing movement.

## Methods

**Participants**. This study involved 15 patients with transient ischemic attack (11 males, 4 females), aged 42 to 76 years (mean = 61.5, standard deviation

(SD) = 9.5). The patients were recruited from the Department of Neurology, Chungnam National University Sejong Hospital from August 2020 to December 2021. The patients with TIA were diagnosed by experienced clinical neurologists within 24 h after symptom onset (mean = 6.0 h, SD = 9.3), which was defined as the point at which the patient reported no longer being in a normal condition. The risk of stroke after TIA was assessed using the ABCD[2] score[77]. The ABCD[2] score is based on five parameters: age, blood pressure, clinical features, duration of TIA, and history of diabetes (mean ABCD[2] score = 4.7, SD = 0.9). All patients received dual antiplatelet therapy with a combination of clopidogrel and aspirin[78]. All patients were scanned within a month after being diagnosed with TIA (mean = 16.2 days, SD = 7.4). This is because (i) patients with TIA have a higher chance of experiencing subsequent stroke within a month[79], and (ii) the study sought to investigate the underlying physiology of this transient physiological state.

Twenty-eight age-matched healthy subjects also participated in this study (mean = 63.8 years, SD = 6.6; 8 males, 20 females). The healthy participants were recruited from the Department of Rehabilitation Medicine, Chungnam National University Sejong Hospital. The healthy subjects had no history of TIA or other neurological diseases within the previous 5 years.

The number of subjects for the patients with TIA, and healthy controls (HC) ($n_{TIA} = 15$, $n_{HC} = 28$) was selected based on sample size estimation studies[60–62] and a power analysis based on the data published in Rehme et al.[27]. The details of the sample size used in this study are addressed in the Discussion section. All subjects were right-handed. The demographic and clinical characteristics of the participants are described in Table 1. The research was approved by the institutional review board of the Chungnam National University Hospital (2019-07-004-001) on 7 August 2019 and was conducted in accordance with the principles embodied in the Declaration of Helsinki and in accordance with local statutory requirements. All participants gave written informed consent to participate in the study.

**Experimental protocol and task paradigm**. The subjects experienced two experimental conditions (right hand (RH) or left hand (LH) movements). More specifically, in each condition, subjects were instructed to close and open their right or left hands synchronized with 1 Hz flickering circle (movement cue). This experimental paradigm for assessing the visually guided motor functions was closely matched to the paradigm established by Grefkes et al.[49]. The experiment comprised 20 s blocks of task interspersed with 20 s blocks of rest, as shown in Fig. 5. Prior to each block of task, an instruction text was shown for 5 s, informing which hand to use in the upcoming block. The sequence of RH and LH blocks was pseudo-randomized. The total scan time was 385 s.

**MRI acquisition**. 7 Tesla MRI (Philips Medical Systems, The Netherlands) was used to obtain all images. BOLD images were scanned using a T2*-weighted gradient echo-planar imaging sequence as follows: repetition time = 2500 ms, echo time = 22 ms, flip angle = 80°, bandwidth = 1619.2 Hz/pixel, field of view = 192 × 192 mm², voxel size = 2 × 2 × 2 mm³, 39 interleaved slices with 0.6 mm slice gap, and number of time frames = 154. Three-dimensional T1-weighted sequence was used to obtain structural images with repetition time = 5.5 ms, echo time = 2.6 ms, flip angle = 7°, field of view = 234 × 234 mm², voxel size = 0.7 × 0.7 × 0.7 mm³, and 269 slices.

**Image pre-processing**. Pre-processing, GLM, and spectral DCM analyses were performed using SPM12 and DCM12.5 software (Wellcome Centre for Human Neuroimaging, UK)[80].

First of all, the unified segmentation algorithm was used to correct the intensity non-uniformity of functional and structural images at 7 T MRI[81]. All the frames were spatially realigned with the mean image to remove the effects of head motion. The realigned functional images were then co-registered with a T1-weighted structural image. The T1-weighted image was segmented into gray matter, white matter (WM), and cerebrospinal fluids (CSF). Both functional and structural

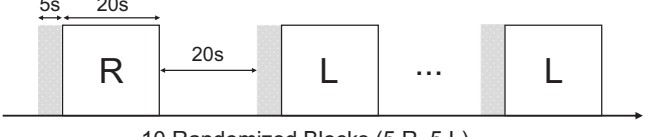

**Fig. 5 Schematic of fist-closing movement task.** The block-design paradigm involved performing right fist-closing (R) or left fist-closing (L) movements according to flickering images. The stimulus was presented randomly in four R and four L block conditions. Each task block was 20 s in duration and separated by a resting baseline of 20 s. Prior to each block of a task, an instruction text was shown for 5 s, informing which hand to use in the upcoming block. The initial resting stabilization of 25 s was also included.

images were spatially normalized into the Montreal Neurological Institute (MNI) space. Finally, the images were spatially smoothed with an 8 mm full-width at a half-maximum Gaussian kernel. Whole images were then divided into the time frames for right-hand movement and left-hand movement with resting state, in order to separate motor task effects for spectral DCM analysis.

**Identification of regions of interest (ROIs)**. A GLM analysis was performed using the canonical hemodynamic response function[82]. The conditions of the fist-closing movement were included as covariates of interest. Parameters for the residual head movement on fMRI images were included in the design matrix of GLM as nuisance variables to mitigate the confounding influence of head motion on the inference of activated voxel location. Very low-frequency noises were suppressed from the BOLD signal using a high-pass filter with a cutoff frequency of 1/128 Hz based on a discrete cosine transform set[80]. GLM was fitted to the individual fMRI signal to estimate the effects of interest for each subject from the first level. A random-effects analysis through summary statistics was then used to analyze the contrast images of all subjects[70]. A second-level one-sample $t$-test design was performed to evaluate the group-level activation during fist-closing movement. The sex difference was included as a nuisance covariate (1 for males, −1 for females).

Based on the GLM result across all participants, ROIs were selected as the nodes for subsequent DCM analysis. This means commonly activated regions in HC and patients with TIA were included in a DCM model, to focus on the characterization of abnormal effective connectivity between groups. This procedure follows the guidelines of previous DCM studies[83,84]. Group-level GLM results are summarized in Table 2, including the significant brain regions, the maximum $t$-value, and the number of voxels in each cluster.

We used empirically identified regions of the SMA, PMC, M1, and IPL in the left and right hemispheres for constructing connectivity models. These ROIs were consistent with the previous DCM studies that used experimental protocol very similar to this study[27,49–51]. Specifically, the ROIs have been selected as core regions of the connectivity model for the motor task—whole-hand fist-closing movements synchronized with the blinks—for the healthy subjects[49,50], and the patients with stroke[27,51]. The SMA was found to be engaged in the preparation and execution of voluntary movements[43]. The PMC was known to play a crucial role in planning and executing goal-orientation reach and grasp actions[54]. The M1 was known as the primary origin of fiber pathways descending to the spinal cord neuron, which ultimately are connected to peripheral muscles[27]. The SMA, M1, and PMC are assumed to be reciprocally connected with each other, as the connections between these regions have all been identified in the macaque brains[49]. Furthermore, the IPL was also included as it was shown to constitute the initial step of the transformation leading from the representation of objects to movement[51,85]. Thus, the IPL in this model can exert influence on premotor regions of the SMA and PMC.

In addition, we used external control ROIs to which the hypothesis (empirical)-based ROI selection can be compared, to confirm whether the connectivity difference between TIA and HC only arises in the motor task-related regions or those connectivity patterns can happen within non-task-specific regions. The external control ROIs were composed of the core regions of the default mode network, including the precuneus (PREC), medial prefrontal cortex (mPFC), and left and right inferior parietal cortex (IPC)[86–88]. These regions have been known to be connected during resting state[86].

**Dynamic causal modeling for task-residual effective connectivity**. Task-residual BOLD time series within the ROIs were extracted by removing task-related

activity within the GLM. The parameters of effective connectivity were then estimated from the task-residual time series using spectral DCM[29,30].

The BOLD time series within each ROI was obtained from each subject using the principal component analysis. Specifically, an 8-mm radius sphere was selected at the center of subject-specific activation peak coordinates within the anatomical mask of ROIs and the nearest local maxima to the group's activation peak coordinates (FWE $p < 0.05$). The anatomical masks for the SMA, M1, IPL, and PMC were extracted based on the Talairach Daemon (TD) database[89,90], using the xjView toolbox (https://www.alivelearn.net/xjview).

Task-residual time series $Y'$ were then calculated by regressing out the task-related signal and systemic confounds from the BOLD time series $Y$:

$$Y' = Y - X\hat{\beta},$$

with

$$X = [x_T, x_M, x_W, x_C, 1],$$

$$\hat{\beta} = (X^T X)^{-1} X^T Y \tag{1}$$

where $x_T$ denotes the task-related regressor modeled by convolving the canonical hemodynamic response function with stimulus function; $x_M$ are vectors of spatial transformation associated with residual head motion effect[91]; $1$ is a vector of ones; and $x_W$ and $x_C$ are nuisance regressors of systemic confounds estimated as the first principal component of voxels within segmented tissue masks of the WM and CSF, respectively[92]. This was based on the assumption that the physiological noise arising from cardiac pulsation and respiration is globally distributed, and neuronal activity-related signals are relatively low in the WH and CSF. Previous studies have shown that the inclusion of these nuisance regressors ($x_M$, $x_W$, $x_C$) corrected the effects of cardiac activity, respiration, and head motion on the inference of functional and effective connectivity[92–94].

We finally applied the spectral DCM[29,30] to the task-residual time series of ROIs using the full connectivity model, as shown in Fig. 6. Specifically, spectral DCM allowed the generation of complex cross spectra among (task-residual) BOLD responses, using a neuronally plausible power-law model and the Fourier transform of the system's Volterra kernels, which are a function of effective connectivity. This spectral DCM did not require the bilinear term of effective connectivity and the spectral contribution term from the experimental input, because task-induced linear effects were already regressed out. Therefore, the number of connectivity models and the inversion process were greatly simplified, which led to an increase in computational efficiency[29,30].

On the other hand, we noted that the task-residual BOLD signal still reflected exogenous neuronal fluctuations. This is supported by several studies showing that its temporal characteristics are synchronized with task engagement[33–36]. In detail, Norman-Haignere et al.[36] assessed the functional connectivity among visual cortex regions using the task-residual BOLD signal. Their results showed that the task-residual connectivity between the fusiform gyrus and the parahippocampal cortex was sensitive to task-related categories, which decreased and increased depending on whether the face and scene tasks were performed, respectively. Using a similar approach, Davies-Thompson et al.[35] found the correlation between task-residual time series of face-selective regions was influenced by task stimulus. The task-residual connectivity between the occipital face area and fusiform face area was increased when the subjects viewed face images compared to other conditions. In the context of task-residual connectivity in regions other than visual cortex areas, Tran et al.[33] have analyzed the task-residual functional connectivity between language and attention network. They observed stronger left-hemisphere laterality —a typical phenomenon of language neural network—among language regions, the inferior frontal gyrus, and the posterior perisylvian region, in task-residual data

**Table 2 Statistical parametric mapping results for all participants (including patients with TIA and HC groups), showing increased brain activation in regions of interest in response to the fist-closing movement (i.e., contrast of right/left-hand movement > rest).**

| Region of interest | Right-hand movement | | | | | Left-hand movement | | | | |
|---|---|---|---|---|---|---|---|---|---|---|
| | $x$ | $y$ | $z$ | Cluster size | $t$-value | $x$ | $y$ | $z$ | Cluster size | $t$-value |
| SMA | −2 | −2 | 64 | 760 | 8.93[a] | 6 | −2 | 58 | 613 | 7.77[a] |
| PMC.R | 52 | 2 | 44 | 124 | 8.15[a] | 52 | 4 | 48 | 251 | 7.60[a] |
| PMC.L | −52 | −2 | 48 | 257 | 7.78[a] | −52 | −2 | 48 | 109 | 7.26[a] |
| M1.R | 34 | −24 | 64 | 7 | 2.92 | 38 | −24 | 52 | 299 | 18.58[a] |
| M1.L | −34 | −24 | 54 | 334 | 13.24[a] | −34 | −24 | 52 | 68 | 3.44 |
| IPL.R | 62 | −34 | 22 | 94 | 8.08[a] | 56 | −34 | 22 | 7 | 6.03[a] |
| IPL.L | −46 | −34 | 22 | 56 | 6.77[a] | −46 | −34 | 24 | 75 | 8.47[a] |

The SPM group results were thresholded at family-wise error rate (FWE) corrected $p < 0.05$ (except for the ipsilateral M1, thresholded at uncorrected $p < 0.005$, $n = 43$). The Montreal neurological institute (MNI) coordinates of the local maxima of the regions, the corresponding $t$-value, and the cluster size were reported. The anatomical labels were determined using the xjView toolbox.
HC healthy controls, TIA transient ischemic attack, ROI region of interest, SMA supplementary motor area, PMC premotor cortex, M1 primary motor cortex, IPL inferior parietal lobule, R right, L left, SPM statistical parametric mapping.
[a]indicates FWE corrected $p < 0.05$.

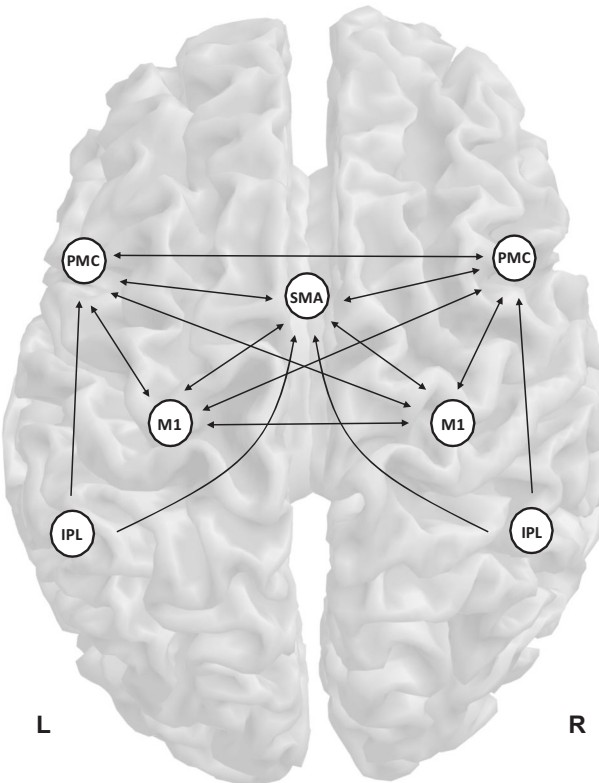

**Fig. 6 Fully connected model for dynamic causal modeling analysis for motor processing data.** The model contains seven regions: the supplementary motor area (SMA), left and right premotor cortex (PMC), primary motor cortex (M1), and inferior parietal lobule (IPL). The black arrow represents effective connectivity. L and R denote left and right, respectively.

compared to resting state. Taken together, these previous results supported that task-residual data are not the same as the resting-state data and could reflect synchronized activity in the context of a task. Therefore, we expected that the Jacobian matrix in task-residual spectral DCM (i.e., A matrix) would represent task-induced neuronal fluctuations, which are different from spectral DCM for resting-state fMRI.

A variational Laplace method was applied to the DCM models considered in this study, to estimate task-residual connectivity parameters, including the posterior distribution and log model evidence for each subject[95]. Parametric empirical Bayes (PEB) was then used to estimate the values and probability distributions of the group-level parameters. The main advantages of using PEB are that it can assess the commonalities and differences of connectivity between subjects at the group level and consider variability in individual connections strength. This approach thus makes individual parameter estimates with higher variance have less influence on group-level results[37]. Using PEB, the parameters of HC and TIA were estimated separately. A contrast of TIA-HC PEB was then performed, using a commonality regressor (1 for both groups) and a contrast regressor (1 for TIA, −1 for HC). Because there was a difference in sex ratio between HC and TIA groups, a sex covariate (1 for male, −1 for female) was also added to exclude any effect of sex on effective connectivity of the within and between group-level PEB.

After estimating the full model, the PEB approach requires performing Bayesian model reduction and Bayesian model average processes. The Bayesian model reduction procedure was used to automatically search over reduced models, which iteratively discards unnecessary parameters for the model evidence, and stops when dropping out any parameters starts to reduce the model evidence. This process allowed comparisons of numerous models quickly and efficiently. The Bayesian model average then calculated the average of the parameters across models, weighted by their evidence[37]. Finally, a threshold based on the Bayesian criterion of posterior probability $p > 0.99$ was used to evaluate the statistical significance of estimated parameters, including task-residual effective connectivity.

The overall procedures for data pre-processing and task-residual DCM are schematically illustrated in Fig. 7.

**Statistics and reproducibility.** The statistical analyses of fMRI data were performed using GLM, spectral DCM, and PEB methods. Technical details with specific parameters are described in the Methods section. The sample sizes used in this study were within a reasonable range of the sample sizes, ensuring 80% power at a 5% significance level. Further details on the sample sizes are addressed in the Discussion section. Additionally, to ensure the reproducibility of the experimental results reported in this study, we have provided the numerical source data—individual connectivity strengths—used for estimating group-level connectivity strengths.

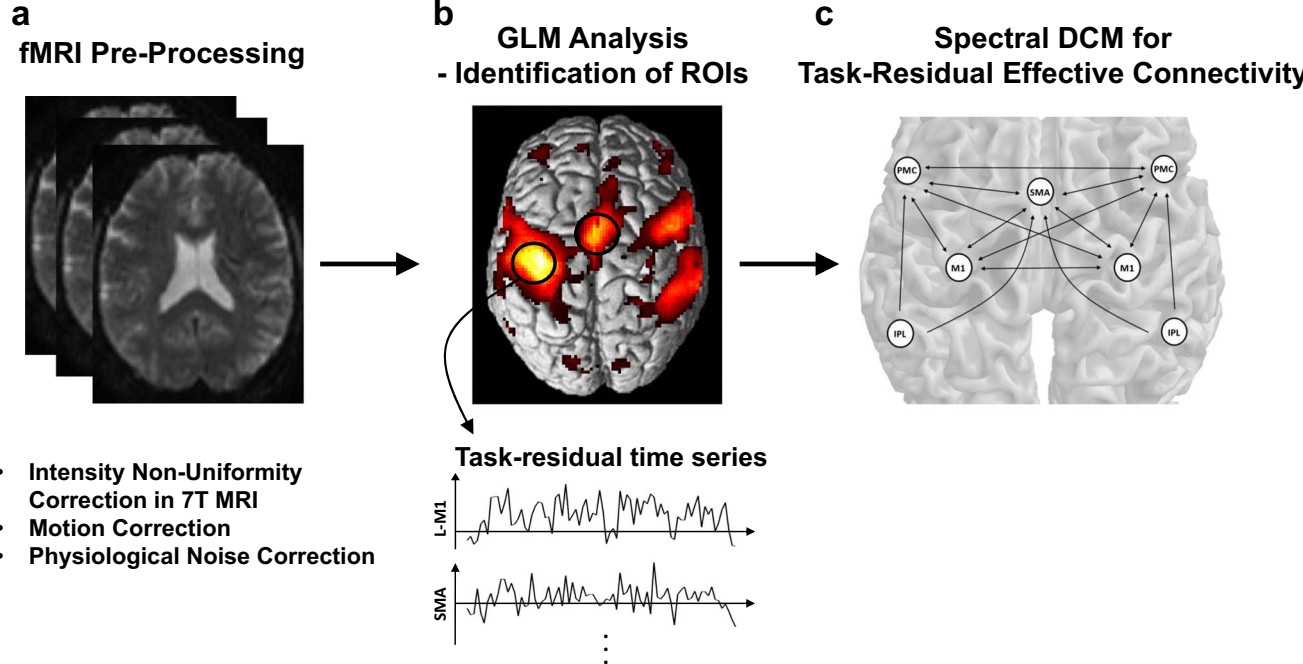

**Fig. 7 Schematic of procedures for functional magnetic resonance imaging (fMRI) data pre-processing and task-residual dynamic causal modeling (DCM) analysis.** Overall procedures are composed of **a** pre-processing of fMRI data, **b** general linear model (GLM) analysis for identifying regions of interest (ROIs), and **c** spectral DCM for making inferences about task-residual effective connectivity.

**Reporting summary**. Further information on research design is available in the Nature Portfolio Reporting Summary linked to this article.

## Data availability

The data generated during and/or analyzed during this study are available from the corresponding author upon request. Numerical source data for Figs. 1 and 2 are provided in Supplementary Data 1. Numerical source data for Figs. 3 and 4 are stored in Supplementary Data 2 and Supplementary Data 3, respectively. Numerical source data for Supplementary Fig. 1 is stored in Supplementary Data 4.

## Code availability

The open-source SPM12, xjView toolbox, and DCM12.5 were the software used to compute regional brain activation, extract the anatomical masks, and estimate effective connectivity from fMRI data. Codes written for extracting task-residual fMRI time series within specified ROIs are available from the corresponding author upon request.

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

## Acknowledgements

This work was supported by a grant of National Research Foundation of Korea (NRF) grant funded by the Korean Government (MSIT) (2019R1C1C1011281, 2022R1F1A1074729), and grants from the Korea Basic Science Institute (D300500, C300300).

## Author contributions

Conceptualization: T.C., I.-Y.J., S.T.; Experimental design and protocol: S.L., I.-Y.J., J.W.S., S.T.; fMRI experiment and participant recruitment: S.L., I.-Y.J., Y.S., H.K., J.W.S., S.T.; Data analysis: T.C., S.T.; Data acquisition: Y.S.; Interpretation of results: T.C., S.L., I.-Y.J., J.W.S., S.T.; Writing manuscript: T.C., S.T.; All authors reviewed the manuscript.

## Competing interests

The authors declare no competing interests.
