## [Peer Review File · Communications Biology]

Reviewers' comments:

Reviewer #1 (Remarks to the Author):

Dear Authors,

Thank you for this manuscript. The technical analysis for the fMRI data and post-processing (including the DCM-analysis) has been well done, nicely described and according to the latest papers within the field. The results have been clearly stated together with figures and tables.

The use of ABCD to assess stroke risk involves a clinical examination of the patient and is such that it could be done even with small or no technical support. The fMRI study together with the DCM analysis is highly technical. Due partly, to the discrepancy in technical support required for the two above mentioned methods, I do not understand why the ABCD score is correlated with the DCM analysis and why it would be interesting.

I would suggest rewriting parts of the method, results and discussion and removing the ABCD score or at least giving it much less significance in the manuscript.

Minor comments: Some of the figures could perhaps be made less complex so that the important findings stand out.

Kind Regards

Reviewer #2 (Remarks to the Author):

The task-based fMRI study investigates the changes in the connectivity of motor-related regions that predict the risk of stroke after a transient ischemic attack. Although the topic is of great interest, I have three major concerns regarding the methodology used. Therefore, I cannot recommend the paper for publication in Communications Biology as it is.

Three major concerns:

- Sample size. The authors address this limitation in the discussion. Still, the argument is not straightforward as illustrated, and even the influential paper they refer to is a matter of debate (e.g. Ingre, 2013, <https://doi.org/10.1016/j.neuroimage.2013.03.03>). Overall, the scientific community is addressing the problem of reproducible fMRI studies, and numerous publications are identifying the sample size as an influential part of the problem to date (e.g. Marek et al. 2022 [10.1038/s41586-022-04492-9](https://doi.org/10.1038/s41586-022-04492-9); Poldrak et al. 2017 [10.1038/nrn.2016.167](https://doi.org/10.1038/nrn.2016.167); Grady et al. 2020 [10.1002/hbm.25217](https://doi.org/10.1002/hbm.25217)). The lack of power analysis and out-of-sample validation (not derived from the leave-one-out) of the robustness of the results are a major concern regarding the generalisability of the findings.

- No control ROI. The authors opted for a hypothesis-based ROIs selection. The rationale is clear and has a solid empirical base. However, the use of external control ROIs could support the hypothesis-based approach.

- The ROIs of interest are involved in functions other than the motor, which aligns with studies that show the involvement of high-order cognitive functions in motor outcomes after a stroke (e.g. Dulyan et al. 2022 [10.1007/s00429-022-02589-5](https://doi.org/10.1007/s00429-022-02589-5)). For instance, SMA is recruited in temporal processing, which has to be maintained during the task assessed in the study (fist-closing at a regular frequency). These additional functions that could predict stroke risk in TIA are mentioned in the discussion. But

little conclusion can be drawn from the current results due to the lack of control neuropsychological assessments and control tasks.

COMMSBIO-23-0556

T. Chu, S. Lee, I.-Y. Jung, Y. Song, H. A. Kim, J. W. Shin, S. Tak

Response to Reviewers

We very much appreciate the reviewers' comments and the opportunity to submit a revised manuscript. Our responses to specific comments of the reviewers are detailed below.

1. Reply to the Reviewer #1

1) *“Dear Authors,*

Thank you for this manuscript. The technical analysis for the fMRI data and post-processing (including the DCM-analysis) has been well done, nicely described and according to the latest papers within the field. The results have been clearly stated together with figures and tables.

The use of ABCD to assess stroke risk involves a clinical examination of the patient and is such that it could be done even with small or no technical support. The fMRI study together with the DCM analysis is highly technical. Due partly, to the discrepancy in technical support required for the two above mentioned methods, I do not understand why the ABCD score is correlated with the DCM analysis and why it would be interesting.

I would suggest rewriting parts of the method, results and discussion and removing the ABCD score or at least giving it much less significance in the manuscript.”

We thank the reviewer for the helpful comment. As suggested, we revised the manuscript to give the ABCD² score much less significance in the manuscript.

Specifically, we removed the contents related to the ABCD² score in the title of the manuscript, and the sections of the abstract, introduction, methods, and results. In the discussion section, we significantly reduced the contents related to the correlation between effective connectivity and ABCD² score. The details of the ABCD² score were moved into the supplementary information.

In addition, we summarized major findings based on the DCM analysis at the end of the discussion section.

2) *“Minor comments: Some of the figures could perhaps be made less complex so that the important findings stand out.*

Kind regards”

Thank you. As suggested, we revised figure 1 and figure 3 to be less complex. For figure 4 and figure 5, we increased the transparency of background-image (brain structure) so that the magnitude of effective connectivity can be identified more clearly.

Figure 1. Schematic of fist-closing movement task. The block-design paradigm involved performing fist-closing movements according to flickering images. The stimulus was presented randomly in four right and four left block conditions. Each task block was 20 seconds in duration and separated by a resting baseline of 20 seconds. Prior to each block of task, an instruction text was shown for 5 seconds, informing which hand to use in the upcoming block. The initial resting stabilization of 25 seconds was also included.

Figure 3. Schematic of procedures for fMRI data pre-processing and task-residual dynamic causal modelling (DCM) analysis.

Figure 4. The posterior mean of the task-residual effective connectivity changes in the TIA and HC participants during fist-closing movements. All the connections presented have significant evidence based on the Bayesian criterion of $p > 0.99$. Black arrows depicted positive influences, while red arrows depicted inhibitory influences. Both groups showed bilateral excitatory connections toward the M1 during right and left hand movements. These results for healthy controls (mean age > 63 years) were in agreement with previous studies, which showed bilateral excitatory patterns of the motor network during hand movements in older adults (Boudrias et al., 2012; Wang et al., 2019). Patients with TIA had greater excitatory connections towards the ipsilateral M1 and higher inhibition connections toward the SMA than healthy controls. Abbreviations: HC, healthy controls; TIA, transient ischemic attack; SMA, supplementary motor area; PMC, premotor cortex; M1, primary motor cortex; IPL, inferior parietal lobule; RHM, right hand movement; LHM, left hand movement.

Figure 5. The significant group differences of the task-residual effective connectivity in response to fist-closing movements. All the connections presented have significant evidence based on the Bayesian criterion of $p > 0.99$. Black arrows depicted stronger connectivity in the TIA group compared to HC, while red arrows depicted weaker connectivity in the TIA group compared to HC. The sum of all connections toward ipsilateral M1 was positively increased in the patient with TIA, compared with the HC during both right and left hand movements. Regarding the SMA region, the effective connectivity to the SMA was decreased in the patients with TIA, compared with the HC during both right and left hand movements. Regarding the PMC regions, the patients with TIA had higher suppressive influence to the PMC during right hand movement, while for the left and movement higher excitatory effects to the PMC were observed from the patients with TIA than the HC. Abbreviations: HC, healthy controls; TIA, transient ischemic attack; SMA, supplementary motor area; PMC, premotor cortex; M1, primary motor cortex; IPL, inferior parietal lobule; RHM, right hand movement; LHM, left hand movement.

2. Reply to the Reviewer #2

- 1) *“The task-based fMRI study investigates the changes in the connectivity of motor-related regions that predict the risk of stroke after a transient ischemic attack. Although the topic is of great interest, I have three major concerns regarding the methodology used. Therefore, I cannot recommend the paper for publication in Communications Biology as it is.*

Three major concerns:

- Sample size. The authors address this limitation in the discussion. Still, the argument is not straightforward as illustrated, and even the influential paper they refer to is a matter of debate (e.g. Ingre, 2013, <https://doi.org/10.1016/j.neuroimage.2013.03.03>). Overall, the scientific community is addressing the problem of reproducible fMRI studies, and numerous publications are identifying the sample size as an influential part of the problem to date (e.g. Marek et al. 2022 10.1038/s41586-022-04492-9; Poldrak et al. 2017 10.1038/nrn.2016.167; Grady et al. 2020 10.1002/hbm.25217). The lack of power analysis and out-of-sample validation (not derived from the leave-one-out) of the robustness of the results are a major concern regarding the generalisability of the findings.”

We thank the reviewer for the helpful comment. In this revision, we carefully reviewed papers the reviewer mentioned (Marek et al., 2022; Poldrak et al., 2017; Grady et al., 2020). We then revised the discussion section, addressing that (1) the sample size of this study ($n_{HC} = 28$, $n_{TIA} = 15$) were relatively small; (2) the sample size could be an influential part of the reproducibility problem, particularly when reporting the “behavior-brain activity relationship” (Marek et al., 2022; Poldrak et al., 2017; Grady et al., 2020). However, neuroimaging-only studies have been still adequately powered with sample sizes in the range of 12 to 25 subjects (Marek et al., 2022; Desmond and Glover 2002; Goulden et al., 2012). This minimum number of samples can be further reduced by increasing the effect size via ultra-high field (e.g., 7T) MRI system (Torrise et al., 2018; Gratton et al., 2022); (3) According to the first reviewer’s suggestion, we significantly reduced the contents related to the behavior-brain activity relationship (i.e., the correlation between effective connectivity and ABCD² score) and the revised manuscript is more focused on the findings only inferred from neuroimaging 7T MRI data – aberrant motor network in patients with TIA. Moreover, we performed a power analysis using DCM connectivity data, where experimental task, ROIs of connectivity model, and participants (stroke patients) were similar with our study (Rehme et al., 2011). This resulted in a required sample size of 14.22 for each group to achieve 80% power at a 5% significance level; (4) Therefore, considering the increase in effect sizes at 7T data and power analysis results, the sample sizes used in this study would fall within the reasonable range of the sample sizes to ensure 80% power at a level of 5% level of significance.

We addressed the issue of sample size in the sections of “2. Methods - 2.1. Participants” and “4. Discussion”:

“2.1. Participants

The number of subjects for the patients with TIA, and healthy controls (HC) ($n_{TIA} = 15$, $n_{HC} = 28$) was selected based on sample size estimation studies (Desmond and Glover, 2002; Goulden et al., 2012; Torrise et al., 2018) and a power analysis based on the data published in Rehme et al., (2011). The details of the sample size used in this study are addressed in the Discussion section.”

“4. Discussion

There are limitations to this study that need to be addressed. First, the group size in this study was relatively small. The numbers of HC and TIA participants were 28 and 15, respectively. Recent studies have identified the sample size as an influential part of the reproducibility problem particularly in the behavior-brain activity relationship estimated by fMRI experiment (Marek et al., 2022; Poldrak et al., 2017; Grady et al., 2020). Marek et al., (2022) quantified this issue by demonstrating that cross-sectional brain-behavior correlations are unreliable without large samples. On the other hand, they also reported that neuroimaging-only studies are typically adequately powered at small sample sizes, and showed that central tendencies of resting-state functional connectivity can be reliably represented averaging within 25 samples. These differences in the statistical power analysis result (Marek et al., 2022) were in line with an opinion of the study (Poldrak et al., 2017): brain activity-behavior correlations inherently have lower power than do average group-activation effects. Main findings of our study were based on the protocols of the neuroimaging-only studies.

In the neuroimaging-only studies, several papers have reported the sample size estimates required to achieve 80% power at a 5% level of significance (Desmond and Glover, 2002; Goulden et al., 2012; Torrisi et al., 2018) and provided guidance on improving power by increasing the effect size (Poldrak et al., 2017; Torrisi et al., 2018; Gratton et al., 2022; Rosenberg et al., 2022). The statistical power is defined as the probability of rejecting the null hypothesis given that the alternative hypothesis is true, and the effect size δ is typically calculated as the absolute difference of the means μ_1, μ_2 of the experimental conditions or independent groups divided by its standard deviation σ : $\delta = \frac{|\mu_2 - \mu_1|}{\sigma}$. For instance, Desmond and Glover (2002) used empirical data acquired from 3T MRI to calculate the effect size based on the percent signal change between conditions and the variability consisting of the intra- and inter-subject standard deviations. These estimated values were then used to create the power curves showing the relationship between the sample size and power. The results showed that 12 subjects are required to ensure 80% power at a single-voxel significance level ($\alpha = 0.05$) and approximately 25 subjects are needed at a more conservative level ($\alpha = 0.000002$). In terms of the DCM-fMRI study, Goulden et al., (2012) performed the power analysis using the typical difference of the means of the effective connectivity between groups and the population standard deviation (the bootstrap distribution) of the parameters. They showed that for the emotion processing and n-back task data, 20 subjects per group is required to achieve 80% power at a significance level of 0.05.

Notably, given the statistical power and significance level, the minimum number of samples (subjects) n can be further reduced by increasing the effect size δ : $n \propto 1/\delta^2$. The effect size is improved by increasing the signal-to-noise ratio (SNR) of the acquired images (Gratton et al., 2022), which can be achieved by using ultra-high field (e.g., 7T) MRI system (van der Zwaag et al., 2009; Torrisi et al., 2018; Hale et al., 2010; Tak et al., 2018). Specifically, compared to 3T, fMRI at 7T allowed for measuring the increased BOLD contrast in the same (motor) region, and increasing the t-statistic value and functional connectivity strength calculated over a common region of interest (van der Zwaag et al., 2009; Hale et al., 2010). In terms of the DCM connectivity, the posterior entropy at 7T was substantially less than that of 3T estimates (Tak et al., 2018). Using these gains at 7T in the effect size, Torrisi et al., (2018) performed power analyses between field strengths and clearly showed that fewer subjects at 7T are necessary to produce comparable effects at 3T.

In this study, we estimated the DCM connectivity strengths from the 7T fMRI images. Therefore, considering (i) the increase in effect sizes at 7T and (ii) power analysis results of previous neuroimaging studies (Desmond and Glover, 2002; Goulden et al., 2012; Torrisi et al., 2018; Marek et al., 2022), the sample sizes used in this study ($n_{TIA} = 28, n_{HC} = 15$) fall

within the reasonable range of the sample sizes to ensure 80% power at a 5% level of significance. We also performed a power analysis using DCM connectivity data obtained from patients with stroke (Rehme et al., 2011). Experimental task and ROIs of connectivity model in Rehme et al., (2011) were similar with our study. In Rehme et al., (2011), mean difference between stroke patients and healthy subjects in effective connectivity among motor areas during fist-closing movements was 0.037. Standard deviations of effective connectivity for stroke and healthy subjects were 0.03 and 0.04, respectively. Using these data, we calculated the sample size required in each group n_i for this study (Lakens 2013; van Belle 2002) and resulted in $n_i = 14.22$ to ensure 80% power ($1 - \beta$) at a 5% significance level (α): $n_i = 2 \left(\frac{Z_{1-\alpha/2} + Z_{1-\beta}}{\delta} \right)^2 = 2 \left(\frac{1.96 + 0.84}{1.05} \right)^2 = 14.22$. Therefore, we can confirm that the number of subjects acquired for each group in this study met the criterion for the minimum sample size that were approximated using power analysis with similar task and patient group dataset. We only reported the posterior means of the connections that were above the Bayesian significance criterion of $p > 0.99$, to confirm the statistical significance of our results. Nevertheless, caution should be taken when generalizing the discussion particularly based on the relationship between connectivity strength and ABCD² score that would be required for larger sample size.”

References:

- Marek et al., 2022. Reproducible brain-wide association studies require thousands of individuals. *Nature* 603, 654–660.
- Poldrack et al., 2017. Scanning the horizon: Towards transparent and reproducible neuroimaging research. *Nat. Rev. Neurosci.* 18, 115–126.
- Grady et al., 2020. Influence of sample size and analytic approach on stability and interpretation of brain-behavior correlations in task-related fMRI data. *Hum. Brain Mapp.* 42, 204–219.
- Desmond and Glover, 2002. Estimating sample size in functional MRI (fMRI) neuroimaging studies: statistical power analyses. *J. Neurosci. Methods.* 118 (2), 115–128.
- Goulden et al., 2012. Sample size estimation for comparing parameters using dynamic causal modeling. *Brain Connect.* 2, 80–90.
- Torrisi et al., 2018. Statistical power comparisons at 3T and 7T with a go / nogo task. *NeuroImage* 175, 100–110.
- Gratton et al., 2022. Brain-behavior correlations: Two paths toward reliability. *Neuron* 110, 1446–1449.
- Rosenberg and Finn, 2022. How to establish robust brain–behavior relationships without thousands of individuals. *Nat. Neurosci.* 25, 835–837.
- Goulden et al., 2012. Sample size estimation for comparing parameters using dynamic causal modeling. *Brain Connect.* 2, 80–90.
- van der Zwaag et al., 2009. fMRI at 1.5, 3 and 7 T: Characterising Bold signal changes. *NeuroImage* 47, 1425–1434.
- Hale et al., 2010. Comparison of functional connectivity in default mode and sensorimotor networks at 3 and 7T. *Magn. Reson. Mater. Phys.*, 23, 339–349.
- Tak et al., 2018. A validation of dynamic causal modelling for 7T fMRI. *J. Neurosci. Methods* 305, 36–45.
- Rehme et al., 2011. Dynamic causal modeling of cortical activity from the acute to the chronic stage after stroke. *Neuroimage* 55 (3), 1147–1158.

- 2) *“- No control ROI. The authors opted for a hypothesis-based ROIs selection. The rationale is clear and has a solid empirical base. However, the use of external control ROIs could support the hypothesis-based approach.”*

We appreciate the helpful comment raised by the reviewer. According to the reviewer’s suggestion, we used external control ROIs composed of the core regions of the default mode network (DMN), to confirm whether the connectivity difference between TIA and HC only arise in the motor task-related regions during the fist-closing movement or those connectivity patterns can happen within non-task-specific regions. The ROIs of the DMN were composed of the precuneus (PREC), medial prefrontal cortex (mPFC), left and right inferior parietal cortex (IPC) (Raichle et al., 2001; Smith et al., 2009; Almgren et al., 2018). These regions have been known to be positively connected during resting state (Raichle et al., 2001). We applied the DCM analysis to the external control ROIs data.

Fig. 6 shows the posterior mean of the task-residual effective connectivity of the external control ROIs in the TIA and HC groups during right and left hand movements. Both groups of the TIA and HC had inhibitory connections within the default mode network during the task state. There was no significant group difference in the effective connectivity among these control ROIs between HC and patients with TIA. These results support that the connectivity difference between TIA and HC during fist-closing movements only arose in the task-specific motor-related regions.

We addressed this issue in the sections of “2. Methods - 2.4.2. Identification of region of interest (ROIs)” and “3. Results – 3.2. Task-residual effective connectivity”.

“2.4.2. Identification of region of interest (ROIs)

In addition, we used external control ROIs to which the hypothesis (empirical)-based ROI selection can be compared, to confirm whether the connectivity difference between TIA and HC only arise in the motor task-related regions or those connectivity patterns can happen within non-task-specific regions. The external control ROIs were composed of the core regions of the default mode network, including the precuneus (PREC), medial prefrontal cortex (mPFC), left and right inferior parietal cortex (IPC) (Raichle et al., 2001; Smith et al., 2009; Almgren et al., 2018). These regions have been known to be connected during resting state (Raichle et al., 2001).”

“3.2. Task-residual effective connectivity

Fig. 6 shows the posterior mean of the task-residual effective connectivity of the external control ROIs in the TIA and HC groups during right and left hand movements. Both groups of the TIA and HC had inhibitory connections within the default mode network during the task state. There was no significant group difference in the effective connectivity among these control ROIs between HC and patients with TIA. These results support that the connectivity difference between TIA and HC during fist-closing movements only arose in the task-specific motor-related regions.”

Figure 6. The posterior mean of the task-residual effective connectivity of the external control ROIs in the TIA and HC groups during right and left hand movements. All the connections represented had significant evidence based on the Bayesian criterion of $p > 0.99$. The results from the TIA and HC groups were similar to each other. There was no significant group difference in the effective connectivity among the control ROIs between HC and patients with TIA. Abbreviations: HC, healthy controls; TIA, transient ischemic attack; mPMC, medial prefrontal cortex; PREC, precuneus; IPC, inferior parietal cortex; RHM, right hand movement; LHM, left hand movement.

References:

- Raichle et al., 2001. A default mode of brain function. *Proc. Natl. Acad. Sci. U.S.A.* 98(2), 676-682.
- Almgren et al., 2018. Variability and reliability of effective connectivity within the core default mode network: A multi-site longitudinal spectral DCM Study. *NeuroImage* 183, 757–768.
- Smith et al., 2009. Correspondence of the brain’s functional architecture during activation and rest. *Proc. Natl. Acad. Sci. U.S.A.*, 106(31), 13040-13045.

- 3) “- The ROIs of interest are involved in functions other than the motor, which aligns with studies that show the involvement of high-order cognitive functions in motor outcomes after a stroke (e.g. Dulyan et al. 2022 10.1007/s00429-022-02589-5). For instance, SMA is recruited in temporal processing, which has to be maintained during the task assessed in the study (fist-closing at a regular frequency). These additional functions that could predict stroke risk in TIA are mentioned in the discussion. But little conclusion can be drawn from the current results due to the lack of control neuropsychological assessments and control tasks.”

We thank the reviewer for the helpful comment. The regions of interest (ROIs) consisted of the primary motor cortex (M1), premotor cortex (PMC), supplementary motor area (SMA), and inferior parietal lobule (IPL), in line with previous studies (Grefkes et al., 2008; Frässle et al., 2015; Rehme et al., 2011; Larsen et al., 2018) that used experimental protocol very similar with this study. Specifically, the ROIs have been selected as core regions of the connectivity model for the motor task – whole-hand fist closing movements synchronized with the blinks – for the healthy subjects (Grefkes et al., 2008; Frässle et al., 2015), and the patients with stroke (Rehme et al., 2011; Larsen et al., 2018).

As the reviewer mentioned, the SMA is recruited in temporal processing of the movement, the preparation and execution of voluntary movements, and is involved in the process of linking (external or internal) stimulus to actual movements (Nachev et al., 2008; Jenkins et al., 2000; Ellermann et al., 1998). Because, as an experimental protocol of this study, the subjects were instructed to close and open their hands synchronized with 1 Hz flickering circle (movement cue), the SMA can be activated during the task and thus was involved in the connectivity model. To validate whether the SMA plays an essential role in the visually guided movement process, we tested 2 models; a fully connected model and one without bidirectional connectivity to the SMA. Bayesian model selection was used to determine the most likely among the candidate models given the observed data. As shown in Fig. 7, the best model was the fully connected model, indicating that the experimental data acquired during fist-closing movements preferred including the SMA in the connectivity model.

In this study, we observed the connectivity towards the SMA was decreased in the patients with TIA compared to the healthy controls. As described in the introduction section, several studies have shown decreased behavioral performance of cognitive and motor functions in the patients with TIA (Simmatis et al., 2017, 2021). Using a robotic exoskeleton system, the patients with TIA were found to have an impairment in visually-guided motor functions after a TIA event. Therefore, the reduced connectivity to the SMA in the patients with TIA could reflect the reduced function of temporal processing during the visually-guided fist closing movement that can be inferred from typical symptoms of the patients with TIA. Similar patterns of the connectivity changes (i.e., decrease in connectivity to the SMA) have been observed during visually cued fist-closing movement from the patients with chronic stroke (Mintzopoulos et al., 2009) and older adults (Boudrias et al., 2012) known to have difficulties in the control of movement, compared with healthy controls.

We addressed the reviewer's comment in the sections "2. Methods – 2.4.2. Identifications of region of interest (ROIs)" and "4. Discussion".

"2.4.2. Identifications of region of interest (ROIs)

We used empirically identified regions of the supplementary motor area (SMA), premotor cortex (PMC), primary motor cortex (M1), and inferior parietal lobule (IPL) in the left and right hemispheres for constructing connectivity models. These ROIs were consistent with the previous DCM studies that used experimental protocol very similar with this study (Grefkes et al., 2008; Frässle et al., 2015; Rehme et al., 2011; Larsen et al., 2018). Specifically, the ROIs have been selected as core regions of the connectivity model for the motor task – whole-hand fist closing movements synchronized with the blinks – for the healthy subjects (Grefkes et al., 2008; Frässle et al., 2015), and the patients with stroke (Rehme et al., 2011; Larsen et al., 2018)."

"4. Discussion"

The connections from other regions to the SMA were found to be reduced in the patients with TIA compared to HC group (from the right IPL and right PMC during right hand movement, the right M1 during left hand movement). The SMA is recruited in the temporal processing of the movement, the preparation and execution of voluntary movements, and is involved in the process of linking (external or internal) stimulus to actual movements (Nachev et al., 2008; Jenkins et al., 2000; Ellermann et al., 1998). The SMA is anatomically connected with the M1 (Johansen-Berg et al., 2004). In a disconnectome study of the patients with stroke, structural disconnection induced by lesion in the medial PMC, the SMA (Goldberg et al., 1985), was shown to be a significant contributor in the prediction of motor functions impairments at

2 weeks and up to 1 year after strokes (Dulyan et al., 2022) Because, as an experimental protocol of this study, the subjects were instructed to close and open their hands synchronized with 1 Hz flickering circle (movement cue), the SMA was activated during the task and thus was involved in the connectivity model in line with previous studies (Grefkes et al., 2008; Frassle et al., 2015; Rehme et al., 2011; Larsen et al., 2018).

We further validated whether the SMA plays an essential role in the fist-closing movement synchronized with the visual movement cue (i.e., relatively simple visually guided movement process), by testing 2 models; a fully connected model and one without bidirectional connectivity to the SMA. Bayesian model selection based on random-effects analysis (Penny et al., 2010) was used to determine the most likely among the candidate models given the observed data. As shown in Fig. 7, the best model was the fully connected model with higher model exceedance probability for both right and left fist closing movements, compared to the model without connectivity to the SMA. This result indicates that the motor network requires the SMA in the process of the fist-closing movement synchronized with the visual movement cue.

In this study, we observed the connectivity towards the SMA was decreased in the patients with TIA compared to the healthy controls. As described in the introduction section, several studies have shown decreased behavioral performance of cognitive and motor functions in the patients with TIA (Simmatis et al., 2017, 2021). Using a robotic exoskeleton system, the patients with TIA were found to have an impairment in visually-guided motor functions after a TIA event. Therefore, the reduced connectivity to the SMA in the patients with TIA could reflect the reduced function of temporal processing during the visually-guided fist closing movement that can be inferred from typical symptoms of the patients with TIA. Similar patterns of the connectivity changes (i.e., decrease in connectivity to the SMA) have been observed during visually cued fist-closing movement in the patients with chronic stroke (Mintzopoulos et al., 2009) and older adults (Boudrias et al., 2012) known to have difficulties in the control of movement, compared with healthy controls.”

Figure 7. Model exceedance probability of the fully connected model (Model 1) and model without bidirectional connection to the SMA (Model 2) across subjects. The best model was the fully connected model for both right and left fist closing movements, compared to the model without connectivity to the SMA. This result indicates that the motor network requires the SMA in the process of the fist-closing movement synchronized with the visual movement cue. Abbreviations: RHM, right hand movement; LHM, left hand movement

References:

- Grefkes, C. et al., 2008. Dynamic intra- and interhemispheric interactions during unilateral and bilateral hand movements assessed with fMRI and DCM. *Neuroimage* 41 (4), 1382–1394.

- Frässle et al., 2015. Test-retest reliability of dynamic causal modeling for fMRI. *Neuroimage*, 117, 56-66.
- Rehme et al., 2011. Dynamic causal modeling of cortical activity from the acute to the chronic stage after stroke. *Neuroimage* 55 (3), 1147–1158.
- Larsen et al., 2018. Modulation of task-related cortical connectivity in the acute and subacute phase after stroke. *Eur. J. Neurosci.* 47 (8), 1024–1032.
- Nachev et al., 2008. Functional role of the supplementary and pre-supplementary motor areas. *Nat. Rev. Neurosci.* 9, 856–869.
- Jenkins et al., 2000. Self-initiated versus externally triggered movements. *Brain* 123, 1216–1228.
- Ellermann et al., 1998. Activation of visuomotor systems during visually guided movements: a functional MRI study. *Journal of Magnetic Resonance*, 131(2), 272-285.
- Simmatis et al., 2017. Robotic exoskeleton assessment of transient ischemic attack. *PLoS One* 12 (12), e0188786.
- Simmatis et al., 2021. Quantifying changes over 1 year in motor and cognitive skill after transient ischemic attack (TIA) using robotics. *Sci. Rep.* 11 (1), 1–12.
- Mintzopoulos et al., 2009. Connectivity alterations assessed by combining fMRI and MR-compatible hand robots in chronic stroke. *Neuroimage* 47.
- Boudrias et al., 2012. Age-related changes in causal interactions between cortical motor regions during hand grip. *Neuroimage* 59 (4), 3398–3405.

REVIEWERS' COMMENTS:

Reviewer #2 (Remarks to the Author):

I would like to commend the authors for the additional analyses and discussion provided. Specifically, the power analysis now shows the suitability of the sample size and the robustness of the results is confirmed by the inclusion of control ROIs. Although other neuropsychological variables are not taken into account, the extension of the discussion to additional higher-order aspects of motor execution as time processing is sufficient for the scope of the study. As a result, the manuscript has improved, and I am happy to advise its publication in Communications Biology.